# Case Study: How Horses Helped a Teenager with Autism Make Friends and Learn How to Work

**DOI:** 10.3390/ijerph16132325

**Published:** 2019-07-01

**Authors:** Temple Grandin

**Affiliations:** Department of Animal Science, Colorado State University, Fort Collins, CO 80523, USA; cheryl.miller@colostate.edu

**Keywords:** autism, social skills, employment, bullying, horses

## Abstract

I was born in 1947 and had autism with speech delay until age four. I am now a college professor of animal science. Horse activities enabled me to make friends through a shared interest in horses. This paper describes the benefits that I experienced from working with horses and my friendships and work skills. A close friendship developed with another student through both riding and horse craft projects. Keeping employment is a serious problem for many people with Autism Spectrum Disorder (ASD). The responsibility of caring for horses and cleaning stalls every day taught me good work skills. My experiences suggest that there were valuable outcomes from working with horses. This may be a beneficial intervention to include in programming for youth with ASD.

## 1. Introduction

When I was three, I had all the signs of severe autism, such as no speech, repetitive behavior and temper tantrums [1,2,3,4]. Autism is a developmental disorder in which there are persistent deficits in social communication, repetitive behavior, restricted interests and either sensory hyper-reactivity or low reactivity to sensory input [5]. Many children with autism have delayed onset of speech [6]. Autism has a wide range of severity, and autistic traits are present in the general population [7]. Some individuals with autism have intellectual talent [8], but other individuals may have much more severe disabilities. Some of the difficulties that students with autism can have are bullying and teasing by classmates [9]. Sensory problems are a core deficit for individuals with autism [10]. Sensory problems may make it difficult for some individuals with autism to tolerate loud noise or scratchy clothes [11,12].

Early intensive speech therapy and teaching turn-taking during games enabled me to become fully verbal at age four. Today, I am a professor of Animal Science at Colorado State University and have had a successful career in the livestock industry for over 40 years [13,14,15,16]. Fortunately, I was mainstreamed into a regular classroom in a small school when I was five. They had older, experienced teachers. The bullies did not start attacking me until high school, because my third-grade teacher, Alice Dietsch, explained to the other children that I had a disability that was not visible in the same way as a wheel chair. Alice Dietsch’s explanation motivated the other students to help me instead of teasing and bullying. The other students encouraged me to play with them and follow the rules of a game. Today, this method is called peer-mediated intervention for autism. Research shows that it is effective [17,18,19].

Enrolling in a large girl’s high school was a complete disaster. I went into puberty at age 14, and this is when bullying started. There was no teacher to explain to the other girls that they should not bully me. This was the time when other girls became interested in clothes, boys, and jewelry; I had zero interest in these things. In elementary school, I had friends through shared interests in art class, sewing, and wood working. In fact, I was the second girl in my fifth-grade elementary school class who was allowed to take wood shop. The girls in the high school were no longer interested in building and making things.

In a larger girl’s high school, I no longer had friends through a special shared interest. Shared interests are an important avenue for teenagers on the autism spectrum to make friends [20,21]. In ninth-grade, at the age of 14, I was kicked out of school for throwing a book at a girl who called me a retard. From there, I went to a special boarding school located on a farm that had both horses and dairy cows. Many of the students at this school had autism. In one of my books, *Animals in Translation* [16], I explained that my life revolved around horses. At that time, I did not realize the full extent of their benefits. This paper will now be divided into two parts: they are how horses helped my social life and friendships, and how they taught me working skills.

## 2. Methods: Horses and My Social Life, and Benefits of Equine Activities

Going to the boarding school did not stop the bullying. The other students called me “tape recorder” because I always used the same phrases. Other students became bored when I kept talking about my favorite topics such as carnival rides [3]. There was one refuge that sheltered me from bullying and teasing: none of the bullies participated in horseback riding. When I was out riding, either on a trail ride or learning equitation in the ring, I was with teenagers who all had the same shared interest in horses. This enabled me to develop a friendship with a girl named Carol. We became roommates and our world revolved around horses. There was one really good horse in our stable that was suitable for showing. Carol and I had to share her when we went to a horse show. I did some classes and she did others. This also taught me how to share.

### 2.1. Horses and a Shared Interest in Craft Projects 

Carol and I did craft and sewing projects that were all about horses. Our prized possessions were our plastic model horses. We made fancy bridles for both the real horses and the model horses. I spent hours making a Western parade bridle and breast collar for both a real horse and a plastic model. Black shoe laces were the perfect size for the bridle and breast collar for a model horse. The silver studs were made from small pieces of tinfoil glued to the shoe laces. In the horse barn hayloft, I found an old carriage harness. I converted parts of the old harness into a Western parade breast collar. The silver studs in the breast collar came from a craft shop, and larger pieces of silver decoration were made from aluminum flashing material that was used for roofing. There were lots of scraps left over from the barn roof project. I had to be ingenious because my budget was extremely limited. Decorative halters were also made by knitting thin strips and hand-sewing them to a halter made from canvas webbing.

An added social benefit was that the head master’s grandchildren loved to play with the plastic horses. When I left the school, I gave the grandkids all my plastic horses along with all my handmade decorations.

Research clearly shows that riding and equine activities improve social communication and engagement [22,23,24,25]. In general, therapy activities involving animals are useful for improving social interaction [26]. Equine activities have the greatest benefits if children keep doing them on a regular schedule [27]. It should be emphasized that either horse or dog-assisted therapy will not work for all individuals on the autism spectrum. I have observed three different ways that individuals with autism respond to animals: they are (1) loving them, (2) being afraid at first then loving them and (3) having sensory problems that will make the child avoid animals [28]. Some examples are as follows: a child fears that the dog will bark or that the horse will whinny and hurt their ears. Smell is another obstacle for some individuals with Autism Spectrum Disorder (ASD).

During my observations of equine therapy for children with autism, there is often not enough emphasis on determining which riders should learn to ride without a side walker. A side walker is a person who holds the horse’s lead rope and leads the horse. There are some individuals who will always need a side walker, but there are others who should become completely independent riders. There is very little literature on determining who needs a side walker and who does not. I observed many situations where people over-accommodate. This is especially a problem with non-verbal individuals.

### 2.2. My Experiences with Modern Equine Therapy

I have observed that the repeated saying of the phrase “good job” can interfere with the flow of an activity. Recently, I assisted with an equine therapy activity with a non-verbal five-year-old. The little girl appeared to become frustrated with a ball-tossing activity, and she threw the ball down. I stepped in and talked to her like a regular child and said, “You chucked it and I cannot get it.” After grabbing the ball, I turned to her and said in a normal voice, “Please catch the ball and throw it gently back to me.” I tossed the ball, she caught it easily, smiled and immediately tossed it back to me. I had to fight the urge to say “Good job.” That would have ruined the flow of the activity. 

At another stable, there was a teenage non-verbal boy, and I think they had underestimated his ability. When I motioned to him to close the sliding stall door, he immediately did it. I treated him as an intelligent person who would understand what I wanted by making a horizontal motion in the direction in which the door would slide. There needs to be more of an attitude of figuring out what an individual can do; often, there is not enough emphasis on developing an individual’s true capabilities.

## 3. Horses and Learning to Work

My social life improved with horses, but I was still a poor student who was not interested in studying. Other than biology class, I had no interest in school work. Mr. Patey, the head master, decided to put me to work caring for the horses and cleaning their stalls. He told my mother that I needed to get through my adolescence, and studying could come later [4]. Every day, I cleaned nine stalls, for eight horses and one donkey. It was hard work, but I thrived on the responsibility of managing the horse barn. In addition to stall cleaning, I put the horses in and out of the farm to pasture and fed them. It was a job, and I did it every day. One of the biggest problems for fully-verbal people diagnosed with autism is employment [29,30]. Taking care of the horses was a job that had responsibility. I was extremely careful never to leave the grain box open. I knew that a horse that gets into an open grain box can over-eat and die.

In the Home Box Office (HBO) movie *Temple Grandin*, one of the horses died. This part in the movie is true. The reason why he died was that my school was short on money and they bought oat straw for the horses to eat instead of hay. I questioned the faculty member who was in charge, saying that I was concerned that the straw was different than regular hay. There was no real hay available, and I was told to feed it to the horses. It ended up causing a deadly case of colic in the largest horse in the barn. I walked and walked him in an attempt to relieve the colic. When the veterinarian arrived, he was appalled that he had been fed oat straw. The next day, they bought real hay and Henry Patey, the head master, made it clear to me that I was not at fault. I was right when I questioned the use of this new coarse-looking straw.

Another benefit of working in the horse barn is that the hard, physical work helped to calm my anxiety. In my experiences with anxiety, Henry Patey let me go for several years without studying, but I was never allowed to become a recluse in my room. Even though I seldom studied, attending classes and meals was required. During my last years at the special school, William Carlock, my science teacher, got me interested in studying by giving me interesting projects. To motivate me to study, studying had to become a pathway to the goal of becoming a scientist.

## 4. Results

Equine activities when I was in high school provided two important benefits. The first was making friends through a joint shared interest in horses and riding. My roommate and I worked hard together to prepare our horses for show. This also taught me the discipline of mastering riding skills. A second benefit was learning responsibility and work skills. I cleaned stalls every day, fed the horses and put them in and out of the barn. This also enabled me to gain greater confidence. The faculty member in charge of the equine program made it clear to me that he recognized my good work in the stables. Equine activities also provided a refuge away from the bullies who tormented me on other parts of the campus.

## 5. Conclusions

In high school, horseback riding provided a refuge from bullying, and I made friends through shared interests in horses. Work skills were learned by caring for horses and cleaning stalls. Learning work skills is important for individuals with autism. Equine activities can benefit youth with ASD in terms of learning both social and working skills.

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
