# Peer review of "Case Study: How Horses Helped a Teenager with Autism Make Friends and Learn How to Work"

_ijerph, 2019, doi:10.3390/ijerph16132325_

Round 1
Reviewer 1 Report
It goes without saying that there will be significant interest in any article published by Temple Grandin and, while Prof. Grandin remains prolific, a reflexive piece such as this wherein Prof. Grandin gives insight into her personal experience of ‘how horses helped a teenager with autism’, as the title has it, will be welcomed by many. As a sociologist, I would be very interested to see what Grandin makes of pieces from other scholars that have considered her relations with animals (e.g. Despret 2016; Grinker 2010) but I appreciate this will not be the case for all readers. Given the nature of the piece, I recommend publication with only minimal changes, which are detailed below.
The term ASD is introduced on p.2L62. A general readership not familiar with autism may not know what this acronym stands for.
P2.L64 – I am unsure what a “side walker” is – may be best to give a few words of explanation.
P2.L65 – “individual” should be plural.
P2 – I’m not sure how the subsection “Horses and a Shared interest in Craft Projects” fits into the section on “Modern Equine Therapy”. The link should either be made clearer or this subsection could be moved, perhaps to the earlier discussion of Carol around P2.L50.
P3 – it was stated previously that the paper will be “divided into two parts” but we here have a third part (Horses and learning to work). Either retitling this section (Conclusion) or editing the previously mentioned sentence will give added clarity.
Despret, V., 2016. What would animals say if we asked the right questions? (Vol. 38). U of Minnesota Press.
Grinker, R.R., 2010. Commentary: On being autistic, and social. Ethos, 38(1), pp.172-178.
Author Response
Dear reveiewer and Journal Editor –
I have made all of the changes that both you and reviewer 1suggested. Eight new references and a paragraph have been added about autism.
The term side walker has also been defined.
The section on horses and shared interests in craft projects has been moved to the section on horses and my social life.
ASD stands for Autism Spectrum Disorder. In the first paragraph, four new references were added to explain autism.
p. 2, line 64 – On line 91, a side walker is explained. It is a person who holds the horse’s lead rope and leads the horse.
p. 2, line 65 – Individuals is corrected on line 91.
p. 2 – Sections sere rearranged per your suggestion.
p. 3 – Sections were rearranged and it now has two sections.
Reviewer 2 Report
This paper looks interesting and attracting to readers in different disciplines. My major comment is no method or results mentioned as normal paper.
Also, the paper should be extended more to give more results and discussion.
I am missing the methodology part.
Author Response
Dear reviewer and Journal Editor:
To respond to the second reviewer’s request for more information on autism and bullying and autism and sensory problems – three references have been added.
As I discussed with you, it is beyond the scope of the paper to do complete literature reviews on these subjects.
Reviewer 3 Report
This manuscript is a rich personal description of a group of factors that acted as barriers and supports to successful participation in important activities for an adolescent with an autism spectrum disorder (ASD). Its value as a scholarly work has the potential to inform interventions to support participation in activities that are important for adult success. The manuscript could be organized in a more scholarly way for maximum impact as a case study. The following are suggestions:
Abstract. Could be all in first person and could explain the following sequence.
Start with a literature review that includes the literature that informs the examples explained in this personal account. You may want to start with the problem areas that might be seen in children, adolescents and adults with ASD
Signs of severe autism (no speech, repetitive behaviors, temper tantrums)
Bullying, negative peer behaviors
Sensory issues
Problems with employment and adult success in ASD
Second part of literature review could talk about supports for ASD that you encountered
Importance of engagement in meaningful activities leads to life satisfaction and a sense of competence, which is essential for psychological and emotional skill development (Law, 2002; Law, Steinwender & LeClair, 1998).
Successes from equine activities and improved social communication and engagement
Benefits of heavy work for anxiety
Maybe how teachers can prepare students to be more supportive of children with ASD if there are any studies on that
Method. What you have written about your overall story and your difficult situations could be described by category in this section. The activities that supported you could be described in Methods under Procedure.
Findings or Results. You could talk about how you responded to the supporting activities that helped you under this section by category.
Discussion. Here, you could talk about why you think the different things worked for you and maybe what it means that therapists, teachers and other professionals should do.
Conclusion. Here, you could reiterated your most important take-away advice such as
Importance of engagement in meaningful activities
Need to be more emphasis on developing individual’s true capacities
Importance of educating classmates about ASD
Since this is a case study, I am not sure if you should talk about your experiences with modern equine therapy. I think it would be better if you can think of examples from your own experience that might help to illustrate these problems
There are a few sections that I am not sure if they contribute to the case study, such as that the horse died from being fed oat straw.
I would be happy to help more with this revision if you want.
Law, M. (2002). Distinguished scholar lecture: Participation in the occupations of everyday life. American Journal of Occupational Therapy, 56(6), 640-649
Law, M., Steinwender, S., & LeClair, L. (1998). Occupation, health and well-being. Canadian Journal of Occupational Therapy, 65(2), 81-91.
Author Response
Thank you.
In general I discussed Reviewer 3’s requests for additional literature review with Dr. Aubrey Fine, who is the editor of the section. When Dr. Fine originally invited me to write this paper, he asked me to write about my personal experiences with horses and not an extensive literature review. Some additional references have been added on autism, bullying, and sesnsory issues.
· Abstract changed to all first person.
· Literature – Four references were added on autism.
· A reference was also added on bullying – reference 9
· Sensory Issues – Three references were added on sensory issues – References 10, 11, and 12.
· Employment – Two references are included and they are references 29 and 30.
· Second Part on additional literature – This goes beyond the scope of my original invited paper.
· Equine activities – There are four references on the benefits of equine activities.
· Method – Recommend not changing the section to Procedure. This is a case study.
· Findings – Recommend keeping present format.
· Discussion – Added a short discussion section. It is limited to horse activities and bullying. A review of all the methods that my mother and teachers used to assist me is beyond the scope of an invited paper on horse activities.
· Conclusion – Made a joint section on Discussion and Conclusions.
· Meaningful activities – Developing true capacities is beyond the scope of a horse paper.
· Modern Equine Therapy – It is important to keep this in because Dr. Fine asked me to write about all my experiences with horses.
· Additional References – I would be happy to co-author with Reviewer 3. The two suggested references go beyond the scope of my original invited paper.
Round 2
Reviewer 2 Report
Well done
Author Response
The abstract was modified per the suggestions from Reviewer 3. Reviewer 3 asked for it to be in the first person.
Reviewer 3 Report
This has improved into a more scientifically written paper, but is still not quite formatted like a regular case study. Also, some language is not quite up to date.
Abstract, second sentence, change to "I am now a college...."
You should add a third sentence in the Abstract that says something like "This paper describes the benefits that I experienced from working with horses on my friendships and work skills."
I think you should delete the sentence that starts with "The students who bullied me..."
You could add a last sentence to Abstract that your experiences suggest important valuable outcomes from working with horses and that this might be a beneficial intervention to include in programming for youth with ASD when possible.
Please change "normal" in line 30 to "regular."
"Disability" is the preferred term instead of "handicap."
I think you should insert "Methods" above the paragraph after your introduction. It should include just the information about the programs that you were in. You can also include the paragraph about what your recent experiences were in this section.
Next, you can add the title, "Results," which can talk about the effects of each of those experiences and go into how well they worked.
The Discussion should include all of your statements about why things worked or did not work and what you think people should learn and do after reading this.
The Conclusion should repeat the most important things about your experiences with the horses and how people should use this information and it should only be a few sentences.
Author Response
The revisions that Reviewer 3 requested.
· Square brackets added
· Conclusions added
· Line 8 – Change sentence to “I am now a college …”
· Line 10 – Added a sentence to the abstract on benefits I experienced from working with horses.
· Line 10 – Deleted sentence on students who bullied me.
· Line 13 – Added a last sentence to the abstract per your recommendation.
· Line 30 – Changed normal to regular
· Line 32 – Changed handicap to disability
· Line 64 – Per your recommendation, Methods was inserted after the introduction.
· Line 150 – Added short results section about the benefits of each horse experience.
· Line 151 – Revised the conclusions